# Fasting blood glucose and risk of incident pancreatic cancer

**Young Jin Kim[1], Chang-Mo Oh[2], Sung Keun Park[3], Ju Young Jung[3], Min-Ho Kim[2,4], Eunhee Ha[5], Do Jin Nam[6], Yeji Kim[6], Eun Hye Yang[6], Hyo Choon Lee[6], Soon Su Shin[7], Jae-Hong Ryoo[8]***

1 Department of Laboratory Medicine, Kyung Hee University Hospital, Kyung Hee University School of Medicine, Seoul, Korea, 2 Departments of Preventive Medicine, School of Medicine, Kyung Hee University, Seoul, Korea, 3 Total Healthcare Center, Kangbuk Samsung Hospital, School of Medicine, Sungkyunkwan University, Seoul, Korea, 4 Informatization Department, Ewha Womans University Seoul Hospital, Seoul, Korea, 5 Department of Occupational and Environment Medicine, College of Medicine, Ewha Womans University, Seoul, Korea, 6 Department of Occupational & Environmental Medicine, Kyung Hee University Hospital, Seoul, Korea, 7 Department of Medicine, Graduate School, Kyung Hee University, Seoul, Korea, 8 Departments of Occupational and Environmental Medicine, School of Medicine, Kyung Hee University, Seoul, Republic of Korea

* armani131@naver.com

**Data Availability Statement:** All relevant data are within the manuscript and its Supporting information files.

**Funding:** This work was supported by the National Research Foundation of Korea in 2020 (grant

## Abstract

### Background

The number of patients with diabetes and impaired fasting blood glucose in Korea is rapidly increasing compared to the past, and other metabolic indicators of population are also changed in recent years. To clarify the mechanism more clearly, we investigated the association between fasting blood glucose and incidence of pancreatic cancer in this retrospective cohort study.

### Methods

In Korea National Health Information Database, 19,050 participants without pancreatic cancer in 2009 were enrolled, and followed up until 2013. We assessed the risk of incident pancreatic cancer according to the quartile groups of fasting blood glucose level (quartile 1: <88 mg/dL, quartile 2: 88–97 mg/dL, quartile 3: 97–109 mg/dL and quartile 4: ≥109 mg/dL). Multivariate Cox-proportional hazard model was used in calculating hazard ratios (HRs) and 95% confidence interval (CI) for incident pancreatic cancer.

### Results

Compared with quartile1 (reference), unadjusted HRs and 95% CI for incident pancreatic cancer significantly increased in order of quartile2 (1.39 [1.01–1.92]), quartile3 (1.50 [1.09–2.07]) and quartile4 (2.18 [1.62–2.95]), and fully adjusted HRs and 95% CI significantly increased from quartile2 (1.47 [1.05–2.04]), quartile3 (1.61 [1.05–2.04]) to quartile4 (2.31 [1.68–3.17]).

number: 2020R1G1A1102257). The funding organization had no role in the design or conduct of this research.

**Competing interests:** The authors have no conflict of interest.

## Conclusion

Fasting blood glucose even with pre-diabetic range was significantly associated with the incident pancreatic cancer in Korean.

## Introduction

Pancreatic cancer burden is increasing globally with population aging, and more cases were diagnosed than past with advancement of diagnostic technology [1]. However, early diagnosis of pancreatic cancer is still challenging due to its clinically silent character until advanced stage, and the small number of patients group makes it difficult to investigate the risk factors or biomarkers [2]. In order to overcome the limitation that only a small number of cases can be studied in a single institution, researches using national wide data were conducted [3–5]. The studies to discover the risk factor for pancreatic cancer found that the life style related risk factors are smoking, and the association of metabolic syndrome such as obesity and diabetes mellitus (DM) [3, 6]. In several studies previously conducted in Korea reported that not only the DM but also the elevated fasting blood glucose was related to risk of pancreatic cancer even if the level was lower than the diagnostic level for DM [5, 7]. The number of patients with diabetes and impaired fasting blood glucose in Korea is rapidly increasing compared to the past [8, 9], and other metabolic indicators of population are also changed in recent years [10].

In this retrospective cohort study, we investigated the association between fasting blood glucose and incidence of pancreatic cancer. We present novel findings on the effects of fasting blood glucose on incidence of pancreatic cancer. Although the importance of fasting blood glucose for pancreatic cancer risk has been suggested [5, 7], this study will contribute with data on which fasting blood glucose is the risk factor for the pancreatic cancer risk, which may be useful in tailoring of prevention strategies and better understanding of mechanisms.

## Materials and methods

### Data sources

The national health insurance system covers the entire population living in South Korea over 97%, suggesting that the database of national health insurance system can be represent the medical service usage of the entire Korean population [11]. In addition, almost Koreans aged more than 40 years are required to undergo a medical health checkup at least once every two years. Information on medical health checkups was collected and stored by the National Health Insurance Corporation (NHIC) in South Korea. In recent years, the national health insurance system in South Korea has provided the sampled database for research purposes after deleting the personal identification information. This sampled database was constructed by stratified random sampling allocated proportionally within each class of age, sex, income level and eligibility status of medical insurance [11]. This sampled database includes information on health checkups linked with the development of pancreatic cancer recorded in Statistics Korea. Ethics approvals for the study protocol and analysis of the data were obtained from the institutional review board of Kyung Hee University Hospital. The informed consent requirement was exempted by the institutional review board because researchers only accessed retrospectively a de-identified database for analysis purposes.

## Study participants and matching

Our study used the data of 223,551 medical checkup participants in 2009 included in the National Health Information Database. Among them, 196 patients previously diagnosed with pancreatic cancer between 2002 and the date before medical checkup in 2009 were excluded. Among the remaining 223,355 participants, 57 participants without information about fasting blood glucose were excluded. Only 381 incident cases of pancreatic cancer developed between 2009 and 2013. Therefore, we adopted matched design to increase the statistical efficiency. 18,669 controls were randomly selected from the participants after 1:50 matching with respect to major covariates (age, gender, body mass index (BMI), systolic blood pressure (BP), total cholesterol, γ-glutamyltransferase (GGT), estimated glomerular filtration rate (eGFR), smoking amount (pack-year), alcohol intake and physical activity). Finally, 19,050 participants were included in final analysis and were monitored for incident pancreatic cancer. The total follow-up period was 82,276.8 person-years and average follow-up period was 4.32 (standard deviation [SD], 0.69) person-years.

## Medical checkup items

The general health check-up of NHIC was conducted thorough 2 stages. The first stage examination is a massive screening test to determine the presence or absence of disease among general population without symptom. The second stage examination is consultation for screening test and more detailed examination to confirm the presence of disease. These health examinations also included a questionnaire for lifestyle or past medical histories. Study data included a physical activity, information provided by a questionnaire, anthropometric measurements and laboratory measurements. Smoking amount was defined as pack-year. The pack-year was calculated from the smoking related questionnaire. Alcohol intake was defined as at least more than 3 times per week. Physical activity was defined as doing the moderate-intensity physical activity at least 30 minutes per day more than 4 days each week or vigorous-intensity physical activity at least 20 minutes per day more than 4 days each week [12]. BMI was calculated as weight (kg) divided by the square of height (meters).

Systolic and diastolic BP was checked by trained examiners. The following laboratory data were checked at the same time that these participants received health examinations: fasting blood glucose, total cholesterol, triglyceride, high-density lipoprotein (HDL)cholesterol, low-density lipoprotein (LDL) cholesterol, serum creatinine (SCr), aspartate aminotransferase (AST), alanine aminotransferase (ALT) and GGT. Kidney function was measured with eGFR, which was calculated using the Chronic Kidney Disease Epidemiology Collaboration (CKD-EPI) equation: eGFR = $141 \times \min (SCr/K, 1)^a \times \max SCr/K, 1)^{-1.209} \times 0.993^{age} \times 1.018$ [if female] $\times 1.159$ [if Black], where SCr is serum creatinine, $K$ is 0.7 for females and 0.9 for males, a is −0.329 for females and −0.411 for males, min indicates the minimum of SCr/$K$ or 1 and max indicates the maximum of SCr/$K$ or 1 [13].

The identification of present baseline DM was based on reviewing data for International Classification of Diseases, 10th Revision, Clinical Modification (ICD-10-CM) code of DM (E10-E14) from 2002 to 2009 years (date of receiving medical health check-up in 2009).

## Outcome definitions

The National Health Insurance database was linked to data of diagnosed disease from Statistics Korea. In this study, the entry date was the first health examination time since 2009 and the last follow-up date for diagnosis of pancreatic cancer was December 31, 2013. Incident pancreatic cancer was verified through detecting participants newly diagnosed with pancreatic cancer during follow-up, based on ICD-10-CM, code C25 (C25.0–25.9), registered in NHIC data. The

primary clinical endpoint of interest for our study was the development of pancreatic cancer as a composite endpoint.

## Statistical analysis

Continuous variables were represented as means ± (standard deviation) or medians (inter-quartile range). Categorical variables were represented as percentage (%). The one-way ANOVA and $X^2$-test were applied to analyze the statistical differences among the characteristics of the participants. The participants were analyzed as quartile groups according to the fasting blood glucose level at the time of enrollment. The person-years were estimated as the sum of follow-up times from the baseline until the diagnosis time of pancreatic cancer development or until the December 31, 2013.

To assess the associations of the quartile groups of baseline fasting blood glucose levels and incident pancreatic cancer, we used Cox proportional hazards models to estimate adjusted hazard ratios (HRs) and 95% confidence intervals (CI) for incident pancreatic cancer. Cox-proportional hazard models were adjusted for the multiple confounding factors. In the multivariate models, we included variables that might confound the relationship between fasting blood glucose and incident pancreatic cancer, which include: age, gender, BMI, systolic BP, total cholesterol, GGT, eGFR, smoking amount (pack-year), alcohol intake and physical activity.

To minimize potential reverse causality, we conducted additional analysis with excluding participants who developed pancreatic cancer within six months or one year of cohort entry, respectively. The results were presented in S1–S5 Tables.

To verify the validity of the Cox-proportional hazard model, the proportional hazard assumption was checked. The proportional hazard assumption was estimated by log-minus-log survival function and found to be graphically unviolated. *P* values <0.05 were considered to be statistically significant. All statistical analyses were performed using SAS (version 9.4, SAS Institute, Cary, NC, USA).

## Results

During follow-up period, 381 incident cases of pancreatic cancer developed between 2009 and 2013. The baseline characteristics of the study participants in relation to the quartile groups of baseline fasting blood glucose levels are shown in Table 1. At baseline, the mean (SD) age and BMI of study participants were 64.5 (10.1) years and 23.9 (3.0) kg/m$^2$, respectively. There were significant differences between all of the listed variables and quartile groups of fasting blood glucose levels. As it was the result after matching, there was no statistically significant difference in all variables except HDL-cholesterol and fasting blood glucose between participant who did not develop pancreatic cancer and participants with pancreatic cancer (S1 Table).

Table 2 shows the hazard ratios and 95% confidence interval for incident pancreatic cancer according to the baseline fasting blood glucose levels. In unadjusted model, the hazard ratios and 95% confidence interval for incident pancreatic cancer comparing the quartile 2,3 and 4 *vs* quartile 1 (reference group) were 1.39 (1.01–1.92), 1.50 (1.09–2.07) and 2.18 (1.62–2.95), respectively (P for trend <0.001).

These associations remained statistically significant, even after further adjustments for covariates in multivariate adjusted model, the adjusted hazard ratios and 95% confidence interval for incident pancreatic cancer were 1.47 (1.05–2.04), 1.61 (1.16–2.23) and 2.31 (1.68–3.17), respectively (P for trend <0.001). The association between fasting blood glucose level and pancreatic cancer was also evaluated using a Kaplan-Meier analysis (S1 Fig). After adding triglyceride to covariates, the hazard ratios and 95% confidence interval for incident pancreatic

**Table 1. Baseline characteristics of participants according to the quartile groups of fasting blood glucose levels (N = 19,050).**

| Characteristic | Overall | Fasting blood glucose level | | | | |
|---|---|---|---|---|---|---|
| | | Quartile 1(<88, n = 4,726) | Quartile 2(≥88, <97, n = 4,961) | Quartile 3(≥97, <109, n = 4,709) | Quartile 4(≥109, n = 4,654) | *P*-for trend* |
| Person-year (total) | 82,276.8 | 20,541.9 | 21,528.3 | 20,417.2 | 19,789.4 | |
| Person-year (average) | 4.32 ± (0.69) | 4.35 ± (0.65) | 4.34 ± (0.65) | 4.33 ± (0.65) | 4.25 ± (0.79) | <0.001 |
| Age (years) | 64.5 ± (10.1) | 64.0 ± (10.5) | 64.1 ± (10.3) | 64.5 ± (10.1) | 65.5 ± (9.6) | <0.001 |
| Gender | | | | | | <0.001 |
| Male (%) | 12,312 (64.6) | 2,823 (59.7) | 3,024 (60.9) | 3,153 (67.0) | 3,312 (71.2) | |
| Female (%) | 6,738 (35.4) | 1,903 (40.3) | 1,937 (39.1) | 1,556 (33.0) | 1,342 (28.8) | |
| BMI (kg/m$^2$) | 23.9 ± (3.0) | 23.5 ± (2.9) | 23.8 ± (2.9) | 24.1 ± (3.0) | 24.5 ± (3.2) | <0.001 |
| Systolic BP (mmHg) | 127.5 ± (15.8) | 125.2 ± (15.6) | 126.5 ± (15.5) | 127.9 ± (15.7) | 130.4 ± (15.9) | <0.001 |
| Diastolic BP (mmHg) | 78.0 ± (9.9) | 76.7 ± (9.8) | 77.6 ± (9.8) | 78.5 ± (9.9) | 79.3 ± (10.0) | <0.001 |
| Total cholesterol (mg/dL) | 195.3 ± (38.0) | 192.4 ± (36.9) | 195.7 ± (36.7) | 197.9 ± (38.0) | 195.2 ± (40.2) | <0.001 |
| Triglyceride (mg/dL) | 120 (85–172) | 109.5 (79–153) | 115 (83–163) | 123 (83–176) | 137 (97–201) | <0.001 |
| HDL-cholesterol (mg/dL) | 54.0 ± (33.5) | 54.7 ± (29.3) | 56.0 ± (38.1) | 55.5 ± (35.4) | 52.9 ± (30.3) | 0.012 |
| LDL-cholesterol (mg/dL) | 113.9 ± (38.3) | 113.7 ± (38.0) | 115.4 ± (37.7) | 115.9 ± (38.1) | 110.7 ± (39.2) | 0.001 |
| SCr (mg/dL) | 1.16 ± (1.39) | 1.10 ± (1.27) | 1.15 ± (1.42) | 1.20 ± (1.49) | 1.20 ± (1.40) | <0.001 |
| eGFR (mL/min per 1.73m$^2$) | 76.3 ± (19.9) | 78.0 ± (19.7) | 76.9 ± (19.5) | 75.8 ± (20.2) | 74.5 ± (20.1) | <0.001 |
| AST (U/L) | 24 (20–30) | 24 (20–29) | 24 (20–29) | 24 (20–30) | 25 (20–33) | <0.001 |
| ALT (U/L) | 21 (16–29) | 19 (15–26) | 20 (15–27) | 21 (16–29) | 24 (17–34) | <0.001 |
| GGT (U/L) | 26 (18–45) | 23 (16–37) | 24 (17–39) | 27 (18–46) | 33 (21–60) | <0.001 |
| Smoking amount (pack-year) | 11.6 ± (18.6) | 10.6 ± (17.8) | 10.2± (17.3) | 12.0 ± (18.9) | 13.8 ± (20.2) | <0.001 |
| Alcohol intake (%) | 18.7 | 14.3 | 16.2 | 20.3 | 24.2 | <0.001 |
| Physical activity (%) | 13.0 | 11.7 | 13.2 | 13.3 | 14.0 | <0.001 |
| Presence of DM (%) | 3914 (20.6) | 513 (10.8) | 564 (11.4) | 751 (15.9) | 2086 (44.8) | <0.001 |
| Development of pancreatic cancer (%) | 381 (2.00) | 63 (1.33) | 92 (1.85) | 94 (2.00) | 132 (2.84) | <0.001 |

Data are means (standard deviation), medians (interquartile range), or percentages.

*P-value by ANOVA-test for continuous variables and Chi square test for categorical variables.

cancer comparing the quartile 2,3 and 4 *vs* quartile 1 (reference group) were 1.45 (1.01–2.08), 1.55 (1.08–2.23) and 2.37 (1.72–3.27) (S2 Table). The adjusted hazard ratios increased in the quartiles 2 and 3 of the models that excluded people who developed pancreatic cancer within six months or a year of entering the cohort (S4 and S5 Tables).

## Discussion

The purpose of this study was to analyze the risk of pancreatic cancer according to the baseline level of fasting glucose in Korean populations. Our data shows that the risk of pancreatic

**Table 2. Hazard ratios (HRs) and 95% confidence intervals (CI) for the incidence of pancreatic cancer according to the quartile groups of fasting blood glucose levels.**

| | Person-years | Incidence cases | Incidence density (per 10,000 person-years) | HR (95% CI) * | |
|---|---|---|---|---|---|
| | | | | Unadjusted | Multivariate adjusted model |
| **Fasting blood glucose levels** | | | | | |
| Quartile 1 | 20,541.9 | 63 | 30.7 | 1.00 (reference) | 1.00 (reference) |
| Quartile 2 | 21,528.3 | 92 | 42.7 | 1.39 (1.01–1.92) | 1.47 (1.05–2.04) |
| Quartile 3 | 20,417.2 | 94 | 46.0 | 1.50 (1.09–2.07) | 1.61 (1.16–2.23) |
| Quartile 4 | 19,789.4 | 132 | 66.7 | 2.18 (1.62–2.95) | 2.31 (1.68–3.17) |
| *P* for trend | | | | <0.001 | <0.001 |
| Age | | | | | 0.998 (0.987–1.010) |
| Gender (female vs male) | | | | | 0.969 (0.752–1.249) |
| BMI | | | | | 0.982 (0.948–1.018) |
| Systolic BP | | | | | 0.999 (0.992–1.006) |
| Total cholesterol | | | | | 0.999 (0.997–1.002) |
| GGT | | | | | 1.000 (0.999–1.001) |
| eGFR | | | | | 1.000 (0.995–1.006) |
| Smoking amount (pack-year) | | | | | 1.000 (0.993–1.006) |
| Alcohol intake | | | | | 0.964 (0.722–1.286) |
| Physical activity | | | | | 0.946 (0.694–1.290) |

Multivariate adjusted model was adjusted for age, gender, BMI, systolic BP, total cholesterol, GGT, eGFR, smoking amount (pack-year), alcohol intake and physical activity.

cancer significantly increased with fasting blood glucose level with pre-diabetic ranges, even after adjusting for multiple covariates.

The hyperglycemia is considered a major factor in cancer development affecting cells by DNA damage, alterations in RNA transcription and oncogenic effect on proteins [14]. The elevated blood glucose has been reported to be associated with various cancers of pancreas, esophagus, liver, colon, cervix, prostate and others [7, 15, 16]. Among the various organs, the pancreas is reported to have a strong positive correlation with elevated blood glucose [7, 15, 17]. The conditions with elevated blood glucose such as diabetes or glucose intolerance is often accompanied by obesity, so the possibility of the effects of BMI on pancreatic cancer incidence may be considered, however elevated blood glucose was a significant indicator of pancreatic cancer incidence even though BMI was controlled in several studies [5, 7, 14] including our study. Another study investigated the correlation between pancreatic ductal adenocarcinoma and hyperglycemia, and they suggested that the hyperglycemia could be a paraneoplastic syndrome [18]. Several studies have also reported that the higher the glucose level, the higher the risk of developing cancers in various organs such as breast, colon, stomach as well as pancreas even if the glucose level is below the level of diabetes diagnosis [7, 17, 19]. There have been reports that this correlation could be explained by hyperinsulinemia or insulin resistance, which can be accompanied by hyperglycemia as well as pre-diabetes [20, 21]. In a meta-analysis, six out of the nine studies reported that the pancreatic cancer rate ratio was significantly higher in the group with the higher blood glucose compared to the group with the lowest blood glucose in the range of blood glucose below 99 mg/dL [20]. In addition, glucose is preferred nutrient for pancreatic cancer cell growth, so the higher the glucose level, the more favorable the environment for cancer cells [21].

Several studies also have been conducted on this relationship in Korea. In 2005, a study in Korea reported a relationship between the risk of pancreatic cancer and blood glucose using

National Health Insurance data from 1992 to 1995 showing that blood glucose range 90–126 mg/dL could be a risk for pancreatic cancer [7]. However, another study using Korean NHS data from 2009–2013 reported that the blood glucose above 100 mg/dL alone was not statistically significant for pancreatic cancer but multiple metabolic components [3]. In another study investigated a relationship between blood glucose and pancreatic cancer incidence in Korean data of same period of 2009–2013, there was a linear relationship after adjusting for age, sex, smoking, alcohol consumption, exercise, and BMI [5]. Accordingly, we needed to clarify the hazard ratio of blood glucose for pancreatic cancer incidence by matching the more disturbance variables that could potentially affect the analysis. As a result, a clearer correlation with dose-response pattern could be found between fasting blood glucose and pancreatic cancer incidence from our study. While the adjusted variables vary from study to study, our study included more variables such as the blood chemistry items, physical measurements and habits and showed clearer trend in hazard ratio.

The merits of the study are medical data including diagnosed pancreatic cancer, behavioral, anthropometric and laboratory measurements verified by Korea national insurance system. Adjustment for baseline potential confounding factors enables us to evaluate the independent effect of fasting blood glucose on incident pancreatic cancer. Nonetheless, limitations of the study should be recognized.

First, we could not check information about family history of pancreatic cancer in our raw data. Family history is looked on as potential risk factors for pancreatic cancer. Thus, unidentified information for family history of pancreatic cancer should be a major limitation for the study.

Second, our results were not acquired from the data designed for research. Because our raw data was clinical records collected from medical exams and related questionnaires, collection bias may exist in our study data. Thus, despite large sample size, collection bias may have an influence on our results.

Third, our study participants are only Korean. Heterogeneity in glucose metabolism can exist among ethnic groups. Thus, further research will be required to determine whether the same outcomes are observed in other ethnic groups.

In conclusion, our study showed the significant relationship between fasting blood glucose and pancreatic cancer incidence clearly in Korean population with pre-diabetic range of fasting blood glucose.

## Supporting information

**S1 Table. Comparison between participants with and without incident pancreatic cancer.**
(DOC)

**S2 Table. Hazard ratios (HRs) and 95% confidence intervals (CI) for the incidence of pancreatic cancer according to the categories of fasting blood glucose in model that including triglyceride as a covariate.**
(DOC)

**S3 Table. Hazard ratios (HRs) and 95% confidence intervals (CI) for the incidence of pancreatic cancer according to the categories of fasting blood glucose.**
(DOC)

**S4 Table. Hazard ratios (HRs) and 95% confidence intervals (CI) for the incidence of pancreatic cancer according to the quartile of fasting blood glucose after excluding the possibility of 6-months reverse causality (N = 19,028).**
(DOC)

**S5 Table. Hazard ratios (HRs) and 95% confidence intervals (CI) for the incidence of pancreatic cancer according to the quartile of fasting blood glucose after excluding the possibility of 1-year reverse causality (N = 18,995).**
(DOC)

**S1 Fig. Kaplan-Meier plot of cumulative incidence of pancreatic cancer according to the quartiles of fasting glucose.**
(TIF)

## Acknowledgments

We used the National Health Insurance Service–National Sample Cohort database and the dataset was obtained from the National Health Insurance Service. Our study findings were not related to the National Health Insurance Service. We specially thanks to researchers who built the database of National Sample Cohort.

## Author Contributions

**Data curation:** Jae-Hong Ryoo.

**Methodology:** Min-Ho Kim, Jae-Hong Ryoo.

**Software:** Chang-Mo Oh, Sung Keun Park, Ju Young Jung, Do Jin Nam.

**Writing – original draft:** Young Jin Kim.

**Writing – review & editing:** Eunhee Ha, Yeji Kim, Eun Hye Yang, Hyo Choon Lee, Soon Su Shin.

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
