## [Decision Letter · Decision Letter 0]

28 Mar 2022

PONE-D-21-35270Fasting blood glucose and risk of incident pancreatic cancerPLOS ONE

Dear Dr. Ryoo,

Thank you for submitting your manuscript to PLOS ONE. After careful consideration, we feel that it has merit but does not fully meet PLOS ONE’s publication criteria as it currently stands. Therefore, we invite you to submit a revised version of the manuscript that addresses the points raised during the review process.

Please submit your revised manuscript within 3 months. If you will need more time than this to complete your revisions, please reply to this message or contact the journal office at plosone@plos.org. Please include the following items when submitting your revised manuscript:A rebuttal letter that responds to each point raised by the academic editor and reviewer(s). You should upload this letter as a separate file labeled 'Response to Reviewers'.A marked-up copy of your manuscript that highlights changes made to the original version. You should upload this as a separate file labeled 'Revised Manuscript with Track Changes'.An unmarked version of your revised paper without tracked changes. You should upload this as a separate file labeled 'Manuscript'.

We look forward to receiving your revised manuscript.

Kind regards,

Altaf Mohammed

Academic Editor

PLOS ONE

Journal Requirements:

2. Please include your tables as part of your main manuscript and remove the individual files. Please note that supplementary tables (should remain/ be uploaded) as separate "supporting information" files

Additional Editor Comments (if provided):

Please carefully address all of the reviewers' comments.

Reviewers' comments:

Reviewer's Responses to Questions

**Comments to the Author**

1. Is the manuscript technically sound, and do the data support the conclusions?

Reviewer #1: Partly

Reviewer #2: Yes

Reviewer #3: Partly

2. Has the statistical analysis been performed appropriately and rigorously? 

Reviewer #1: No

Reviewer #2: Yes

Reviewer #3: I Don't Know

3. Have the authors made all data underlying the findings in their manuscript fully available?

Reviewer #1: Yes

Reviewer #2: Yes

Reviewer #3: Yes

4. Is the manuscript presented in an intelligible fashion and written in standard English?

Reviewer #1: No

Reviewer #2: Yes

Reviewer #3: Yes

5. Review Comments to the Author

Reviewer #1: In this article, Kim et al examined the association of fasting blood glucose level and incident pancreatic cancer risk using Korea National health Information Database. The study included 19,050 participants at the baseline. 381 incident cases were observed during an average of 4.32 years follow-up. The authors found the increase of blood glucose level was associated with pancreatic cancer in each quartile, even in the pre diabetic range.

There are several concerns of the study.

1. The authors indicated that the dataset was sampled and represented a portion of Korea National health Information Database. It would be useful to describe how the sampling was done and if it is representative of the population.

2. It is unclear why authors chose the case-control study design with 1:50 matching controls. Why not the entire dataset? Also In page 13, Result section, “During 82,276.8 person-years of follow-up, 381 (2.00 %) incident cases of pancreatic cancer developed between 2009 and 2013.” This is misleading because the authors chose 1:50 matching, but not 2% of incident rate.

3. It is surprising to see the multivariate adjusted models had much higher HR than the unadjusted models, although the samples were matched for the covariates.

4. Quartiles of glucose levels were examined in the study. It would be interesting to see the group with >126 mg/dL. It would be also interesting to examine if there is any linear association between glucose level and pancreatic cancer.

5. Does the study account for the effect of diabetic medication?

6. In the results section, it is unclear “As expected, all variables were statistically insignificant except HDL-cholesterol and fasting blood glucose, as it was the result after matching”. Does it mean that they are not significant between cases and controls? If so, please indicate it.

7. Please provide a Kaplan-Meier plot to show the incident rate for different quartiles

8. Proofreading would be needed to correct typos and grammatical errors.

Reviewer #2: Authors are carried out retrospective analysis of patient visits to the fasting glucose levels vs pancreatic cancer incident rates. Overall authors used large pool of Korean patients with fasting glucose levels ranging <88 to >109 and patients with higher quartiles of fasting glucose significantly associated with pancreatic incidence.

Minor Comments.

1. Authors should comment on quartile 4 have a higher triglycerides - thus any effect on the pancreatic cancer incidence as independent of fasting glucose levels?

2. Is their any difference in BMI of Male vs Female?

3. Smoking is higher in the Quartile 4- does authors analyzed any effect on the pancreatic cancer

Reviewer #3: Early detection of pancreatic cancer is challenging due to clinically silent progression of the disease until it reaches an advanced stage. It is important to understand and identify modifiable risk factors as it can help develop strategies to mitigate the risk of cancer. There are some known lifestyle-related risk factors such as alcohol and tobacco smoking, and metabolic syndrome such as obesity and diabetes mellitus (DM). The purpose of the submitted retrospective cohort study was to analyze the risk of pancreatic cancer according to the baseline level of fasting glucose in the South Korean population for which they used the National Health Insurance Corporation (NHIC) database that covers 97% of the population. They obtained data from 223,551 medical checkup participants included in NHIC database in 2009 and excluded those with previous pancreatic cancer diagnosis or with no fasting glucose data. The analysis included a total of 381 incident cases of pancreatic cancer that developed between 2009 and 2013, and 18,669 controls that were randomly selected from the participants after 1:50 matching for covariates resulting in a total of 19,050 participants. The participants were analyzed as quartile groups according to the fasting blood glucose level at the time of enrollment (quartile 1: <88 mg/dL, quartile 2: 88�97 mg/dL, quartile 3: 97�109 mg/dL and quartile 4: ≥109 mg/dL), and multivariate Cox proportional hazards models were used to estimate adjusted HRs and 95% CIs for incident pancreatic cancer. The results showed that that the risk of pancreatic cancer increased statistically significantly with fasting blood glucose level (quartile 1 [reference] vs. quartiles 2, 3 and 4) with prediabetic ranges, even after adjusting for multiple covariates. The authors mentioned that this study adds value to the existing literature by providing a clearer correlation between fasting blood glucose levels and pancreatic cancer incidence.

Several studies have already been conducted in Korea and elsewhere reporting the association of not only DM but also of elevated fasting blood glucose with increased risk of pancreatic cancer. A 2019 study by Koo et al. (reference 5) used the same South Korean national database as the current manuscript to report HRs for fasting glucose level and pancreatic cancer incidence. They reported a linear relationship between fasting glucose levels and pancreatic cancer incidence where the cumulative incidence rate of pancreatic cancer increased statistically significantly with elevation in fasting glucose level, not only in those with DM but also in those with prediabetes or high normal range of fasting blood glucose levels, and even after adjustment of well-known risk factors. Therefore, the reviewer’s enthusiasm has been substantially lowered by the lack of novelty in the submitted manuscript. In addition, this reviewer also thought the study could have explored areas that other studies based on the Korean population may not have explored, for example, include data from a longer period and analyze longitudinal trajectory of fasting glucose levels, assuming that longitudinal glucose measurements are available in the database as the population undergoes check up every two years, as this could have added a new dimension to the analysis. The reason for quartile 4 threshold is not clear, and why the quartiles are not distinct for ≥109 � <126 mg/dL and ≥126 mg/dL? The reviewer was unable to find any information on anti-diabetes medication and the multivariate analysis accounted for age, gender, BMI, systolic BP, total cholesterol, GGT, eGFR, smoking amount (pack-year), alcohol intake and physical activity. Furthermore, many patients may have had occult/subclinical pancreatic cancer where reverse causality could be the reason for increased glucose level and although it was not possible to know that in this retrospective setting, an assessment of the duration of hyperglycemia and longitudinal data on glucose levels would have been useful. Thanks to the authors for acknowledging some limitations of their study. This reviewer thinks HbA1c levels were not available in the NHIC database, which is why the study was unable to include the data. The manuscript is well-written overall and easy to read.

6. PLOS authors have the option to publish the peer review history of their article (what does this mean?). If published, this will include your full peer review and any attached files.

---

## [Author Response · Author response to Decision Letter 0]

27 Jul 2022

We are grateful for editor and reviewer thorough consideration and scrutiny of our manuscript, “Fasting blood glucose and risk of incident pancreatic cancer". We acknowledge that the quality of our manuscript was improved by the scrutinizing efforts of the reviewers and editors. As your suggestion, we performed additional analyses and revised manuscript. The file titled "Response to Reviewers" contains a detailed list of every change made to the revised manuscript.

---

## [Decision Letter · Decision Letter 1]

19 Aug 2022

PONE-D-21-35270R1Fasting blood glucose and risk of incident pancreatic cancerPLOS ONE

Dear Dr. Ryoo,

Thank you for submitting your manuscript to PLOS ONE. After careful consideration, we feel that it has merit but does not fully meet PLOS ONE’s publication criteria as it currently stands. Therefore, we invite you to submit a revised version of the manuscript that addresses the points raised during the review process. Please submit your revised manuscript by Oct 03 2022 11:59PM. If you will need more time than this to complete your revisions, please reply to this message or contact the journal office at plosone@plos.org. Please include the following items when submitting your revised manuscript:A rebuttal letter that responds to each point raised by the academic editor and reviewer(s). You should upload this letter as a separate file labeled 'Response to Reviewers'.A marked-up copy of your manuscript that highlights changes made to the original version. You should upload this as a separate file labeled 'Revised Manuscript with Track Changes'.An unmarked version of your revised paper without tracked changes. You should upload this as a separate file labeled 'Manuscript'.If applicable, we recommend that you deposit your laboratory protocols in protocols.io to enhance the reproducibility of your results. Protocols.io assigns your protocol its own identifier (DOI) so that it can be cited independently in the future. For instructions see: https://journals.plos.org/plosone/s/submission-guidelines#loc-laboratory-protocols. Additionally, PLOS ONE offers an option for publishing peer-reviewed Lab Protocol articles, which describe protocols hosted on protocols.io. Read more information on sharing protocols at https://plos.org/protocols?utm_medium=editorial-email&utm_source=authorletters&utm_campaign=protocols.

We look forward to receiving your revised manuscript.

Kind regards,

Altaf Mohammed

Academic Editor

PLOS ONE

Journal Requirements:

Additional Editor Comments:

Please make sure to include all the additional figures (example: Kaplan-Meir plot,..) or tables (example: Table referred under Reviewer 2's comment#1) in the manuscript or the supplemental data.

Reviewers' comments:

Reviewer's Responses to Questions

**Comments to the Author**

1. If the authors have adequately addressed your comments raised in a previous round of review and you feel that this manuscript is now acceptable for publication, you may indicate that here to bypass the “Comments to the Author” section, enter your conflict of interest statement in the “Confidential to Editor” section, and submit your "Accept" recommendation.

Reviewer #1: All comments have been addressed

Reviewer #2: All comments have been addressed

Reviewer #4: All comments have been addressed

2. Is the manuscript technically sound, and do the data support the conclusions?

Reviewer #1: Yes

Reviewer #2: Yes

Reviewer #4: Yes

3. Has the statistical analysis been performed appropriately and rigorously? 

Reviewer #1: Yes

Reviewer #2: Yes

Reviewer #4: Yes

4. Have the authors made all data underlying the findings in their manuscript fully available?

Reviewer #1: Yes

Reviewer #2: Yes

Reviewer #4: Yes

5. Is the manuscript presented in an intelligible fashion and written in standard English?

Reviewer #1: Yes

Reviewer #2: Yes

Reviewer #4: Yes

6. Review Comments to the Author

Reviewer #1: (No Response)

Reviewer #2: Authors incorporated all the responses in the revised manuscript. No additional comments from this reviewer

Reviewer #4: Early detection of pancreatic cancer is an important area of research which needs to be developed to catch the disease at the early stage to improve patient survival. This article is interesting as authors have investigated the association between fasting blood glucose and incidence of pancreatic cancer in a retrospective cohort study. From this study authors concluded that Fasting blood glucose even with pre-diabetic range was significantly associated with the incident pancreatic cancer in Korean. Authors have addressed all questions raised by the reviewers satisfactorily.

7. PLOS authors have the option to publish the peer review history of their article (what does this mean?). If published, this will include your full peer review and any attached files.

Reviewer #1: No

Reviewer #2: **Yes: **Chinthalapally V. Rao

Reviewer #4: No

---

## [Author Response · Author response to Decision Letter 1]

22 Aug 2022

We included all additional figures or tables made during the review process in the manuscript or supplementary material. We indicated where we made changes in response to the suggestions of reviewer colored in red color. Thank you very much for us to have a precious opportunity to revise the manuscript once again

---

## [Editor Report · Decision Letter 2]

24 Aug 2022

Fasting blood glucose and risk of incident pancreatic cancer

PONE-D-21-35270R2

Dear Dr. Ryoo,

We’re pleased to inform you that your manuscript has been judged scientifically suitable for publication and will be formally accepted for publication once it meets all outstanding technical requirements.

Kind regards,

Altaf Mohammed

Academic Editor

PLOS ONE
---

## [Editor Report · Acceptance letter]

18 Oct 2022

PONE-D-21-35270R2 

Fasting blood glucose and risk of incident pancreatic cancer 

Dear Dr. Ryoo:

I'm pleased to inform you that your manuscript has been deemed suitable for publication in PLOS ONE. Congratulations! Your manuscript is now with our production department. 

Kind regards, 

on behalf of

Dr. Altaf Mohammed 

Academic Editor

PLOS ONE